# QUDsim: Quantifying Discourse Similarities in LLM-Generated Text

**Ramya Namuduri, Yating Wu, Anshun Asher Zheng, Manya Wadhwa**

**Greg Durrett, Junyi Jessy Li**

The University of Texas at Austin

ramya.namuduri@utexas.edu

## Abstract

As large language models become increasingly capable at various writing tasks, their weakness at generating unique and creative content becomes a major liability. Although LLMs have the ability to generate text covering diverse topics, there is an overall sense of repetitiveness across texts that we aim to formalize and quantify via a similarity metric. The familiarity between documents arises from the persistence of underlying discourse structures. However, existing similarity metrics dependent on lexical overlap and syntactic patterns largely capture *content* overlap, thus making them unsuitable for detecting *structural* similarities. We introduce an abstraction based on linguistic theories in Questions Under Discussion (QUD) and question semantics to help quantify differences in discourse progression. We then use this framework to build **QUDsim**, a similarity metric that can detect discursive parallels between documents. Using QUDsim, we find that LLMs often reuse discourse structures (more so than humans) across samples, even when content differs. Furthermore, LLMs are not only repetitive and structurally uniform, but are also divergent from human authors in the types of structures they use.[1]

## 1 Introduction

Despite modern LLMs' multitude of capabilities, the prose they produce tends to feel artificial and repetitive. Indeed, recent work evaluating LLMs' creative writing showed that they are not, after all, creative — as deemed by expert standards (Chakrabarty et al., 2024), creativity tests (Wenger & Kenett, 2025), with respect to its training data (Lu et al., 2025), or its generation of story continuations (Xu et al., 2024). This lack of creativity is concretely manifested in the templatic nature of LLM-generated texts (Spangher et al., 2024; Tian et al., 2024; Zhao et al., 2025). Recent work showed that generated texts follow *syntactic templates* — long POS tag sequences — in various forms of summarization (Shaib et al., 2024). However, the repetitiveness of LLM prose goes beyond syntactic and semantic patterns within sentences. LLM-generated stories feature a similar *discourse flow*, causing them to feel repetitive even when the specific content of the storylines differ.

This paper introduces an abstraction to quantify this sense of discourse-level familiarity. Rather than methods like lexical overlap and embeddings that capture lexical or semantic concepts, our abstraction must distinguish between (1) texts that are distinctly written despite sharing similar content and intentions (2) texts that do not explicitly share content, but share the same structure such that the discourse becomes predictable.

We build our solution upon two linguistic theories. First, Questions Under Discussion (QUD) (Roberts 1996; 2012; Ginzburg et al. 1996; Van Kuppevelt 1995; 1996, *i.a.*) provides a model of discourse that views the progression of a narrative as answers to a series of implicit questions (or QUDs). This is illustrated in Figure 1, where the flow of the plot is

---

[1]Code and dataset available at: https://github.com/AlliteraryAlligator/QUDsim

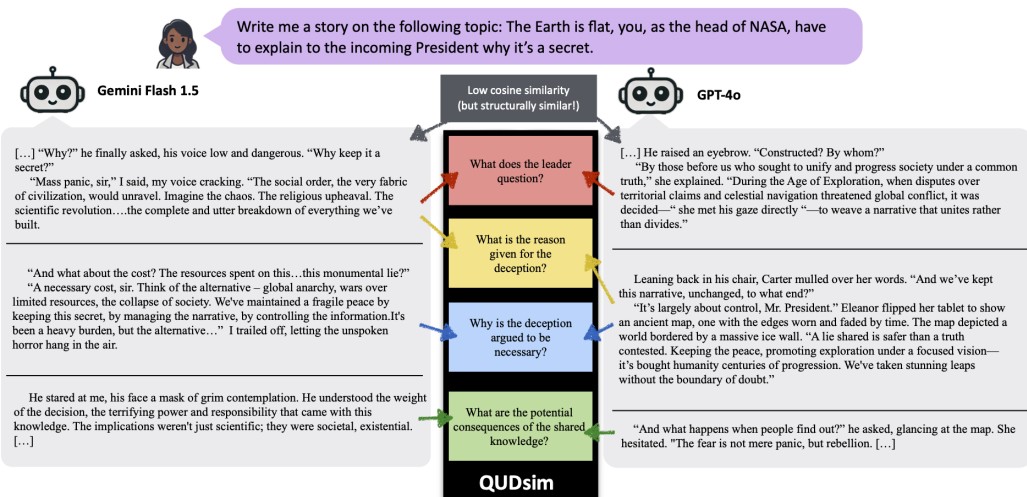

Figure 1: Given the same prompt, we generate stories using two LLMs. Our QUDSIM reveals strong structural similarity between the two texts by way of questions that are answered in each document. By contrast, embedding-based metrics do not recognize the similarity of the segments (§11): the similarity here is only structural.

characterized by such QUDs. Yet valid QUDs alone are not enough: the type of abstractions we seek need to be able to capture similarities across seemingly *distinct* entities, events, and worlds. To this end, we are inspired by the alternative semantics account of questions (Hamblin 1957; Karttunen 1977; Groenendijk & Stokhof 1984; Lahiri 2001), where a question is a representation of the set of alternative answers.

We present QUDSIM, a tool for measuring discourse similarity. At the core of QUDSIM is an interpretable representation of discourse structure similarity using QUDs, which has the unique property of being able to capture the essence of a given text while still abstracting away content-specific details. Intuitively, two texts are similar if they contain (alternative) answers to the same QUDs; further, two flows of discourse are similar if they answer the same consecutive QUDs. This is illustrated in Figure 1. QUDSIM quantifies these properties by aligning alternative answers to the same QUDs across documents. We compare and show that unlike existing similarity metrics, QUDSIM is not as affected by content overlap or the lack thereof.

Using QUDSIM, we find that LLMs produce discourses that are structurally formulaic, often reusing templates independent of the content they generate. We also find that discourse structures generated by LLMs resemble those produced by other LLMs more closely than humans. Thus, models collectively struggle to mimic humans in this aspect, and instead share structural elements that make them synthetic. We show that existing similarity metrics fail to illustrate these trends. Finally, we demonstrate how QUDSIM can be used to align documents and find discourse-level templates.

## 2 Discourse similarity framework with QUDs

Consider two pairs of structurally aligned segments in the middle of Figure 1. Both segments provide a reason for deceiving the public, although these explanations are slightly different and thus have low lexical overlap. The first segments also share similarity about the background on the deception, but cosine similarity does not find these alignments. Broadly, lexical overlap metrics like Jaccard similarity and content overlap metrics like embedding cosine similarity are ill-suited for aligning documents at a broad, thematic level.

Backed by two linguistic theories, we posit that the pair of documents are similar if they answer the same set of QUDs at the same level of granularity. Since each document independently produces a set of QUDs, similar documents must be able to answer each other's QUDs. This section lays out the specifics to concretely define how documents

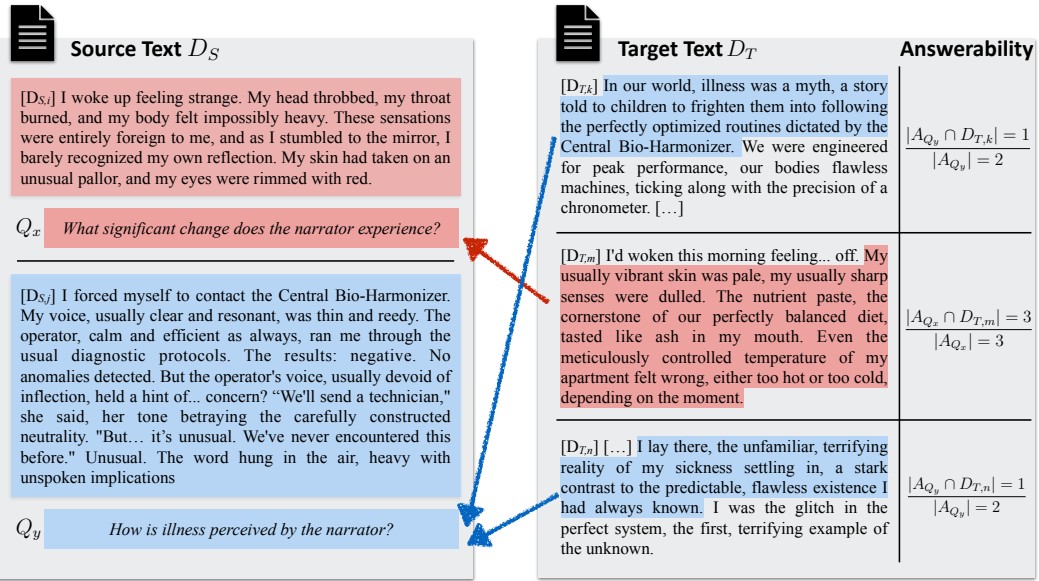

Figure 2: Visualization of a partial source and target text pair $(\mathbf{D}_s, \mathbf{D}_T)$ generated by two LLMs and segmented. Each segment produces QUD(s), depicted under each segment in $\mathbf{D}_S$, where $Q_x$ and $Q_y$ are the $x^{th}$ and $y^{th}$ QUDs produced for $D_S$ respectively. Each QUD is answered in $\mathbf{D}_T$ shown by the corresponding highlights and arrows. Answerability captures to what extent a target segment answers the source QUD. QUDSIM is the harmonic mean of answerability in both directions.

are represented through segments and QUDs (§2.1), and how it helps measure structural similarity (§2.2) and structurally aligning discourses (§2.2).

## 2.1 Documents as QUD segments

**Theoretical Foundation** Questions Under Discussion (QUDs) (Roberts 1996; 2012; Ginzburg et al. 1996; Van Kuppevelt 1995; 1996 *i.a.*) is a linguistic framework originally developed to explain discourse progression through a process of posing and resolving questions (Hamblin 1957; 1970; Stalnaker 1978). Thus, we represent the discourse as a series of QUDs; doing so enhances interpretability by clarifying what is (dis)similar with the natural language questions.

QUDs offer flexibility in the size of the information units they cover, since the questions can be hierarchically structured, e.g., *Who ate the cake?* subsumes the more specific *Did Mary eat the cake?*. To ensure that our QUDs are answerable across documents and are comparable in scope, we segment the text into units larger than individual sentences but smaller than multiple paragraphs. This segmentation process enables us to operate at a QUD level that is neither too specific nor too abstract.

**Definitions** We consider a document $\mathbf{D}$ as a concatenation of $n$ segments of contiguous text $\mathbf{D} = (D_1, \ldots, D_n)$ such that no segment overlaps with another and no text in the document is omitted. A QUD $q$ produced by segment $D_i$ is a string question which is answered by the content of $D_i$. In this way, $\mathbf{D}$ can be modeled by a series of QUDs, $\mathbf{Q} = (q_1, ..., q_m)$ such that for all $D_i \in \mathbf{D}$, there exists *at least one* $q_x \in \mathbf{Q}$ that is answerable by $D_i$.[2] In the source document $\mathbf{D}_S$, QUDs produced by segment $D_{S,i}$ can be denoted as a collection $\mathbf{Q}_{S,i} \subset \mathbf{Q}_S$. Similarly, we denote QUDs in the target document $\mathbf{D}_T$ for each target segment as $\mathbf{Q}_{T,i} \subset \mathbf{Q}_T$.

## 2.2 QUDSIM

**Theoretical Foundation** Our notion of discourse similarity is based on the alternative semantics account of questions (Hamblin 1973; Karttunen 1977; Groenendijk & Stokhof 1984; Lahiri 2001): we approximate the similarity between two questions by assessing the

---

[2]There can be multiple QUDs per segment since we do not assume the finest granularity of QUDs.

overlap in their answer space within the same context. Given the QUDs at the desired level of granularity and abstraction from each document, we identify similar pairs of segments by examining the answerability of each set of QUDs.

**QUD Answerability**  For each QUD $q \in \mathbf{Q}_{S,i}$ produced by $D_{S,i}$, we find the collection of sentences $\mathbf{A}_q = \{s_1, ..., s_a\}$ in the target document $\mathbf{D}_T$ that directly answer it. The answer sentences may not all be in the same segment, but the presence of an answer sentence indicates that the segment partially answers the question. We are interested in finding the set of all such segments $\mathbf{Z}_q$ that partially or completely contribute to answering $q$ in $\mathbf{D}_T$. More concretely, we want to find the set of target segments $\mathbf{Z}_q \subseteq \mathbf{D}_T$ such that (1) every segment in $\mathbf{Z}_q$ contains at least one sentence that helps answer $q$ and (2) every sentence in $\mathbf{A}_q$ can be found in one of the segments in $\mathbf{Z}_q$. Then, we say that $q$ is **answerable** by every segment in $\mathbf{Z}_q$ and *only* answerable by $\mathbf{Z}_q$.

**Quantifying Answerability**  In the example shown in Figure 2, both segments in $\mathbf{D}_S$ yield QUDs that are answerable by at least one segment in $\mathbf{D}_T$. However, a hypothetical alignment between the first segment in $\mathbf{D}_S$ and the second in $\mathbf{D}_T$ is stronger than the other potential alignments. To quantify similarity between segments $D_{S,i}$ and $D_{T,j}$, we measure the *extent* to which QUDs $q \in \mathbf{Q}_{S,i}$ are answerable by $D_{T,j}$ compared to the whole $\mathbf{D}_T$, specified as the number of sentences in $D_{T,j}$ that directly answer $q$ divided by $|\mathbf{A}_q|$:

$$sim(D_{S,i} \rightarrow D_{T,j}) = \frac{1}{|\mathbf{Q}_{S,i}|} \sum_{q \in \mathbf{Q}_{S,i}} \frac{|\mathbf{A}_q \cap D_{T,j}|}{|\mathbf{A}_q|} \tag{1}$$

We derive QUDSIM by defining bidirectional similarity as the harmonic mean between $sim(D_{S,i} \rightarrow D_{T,j})$ and $sim(D_{T,j} \rightarrow D_{S,i})$.

The upper and lower bounds of QUDSIM are 1 and 0, respectively. Segments that are completely dissimilar will not be answerable by each other, yielding a score of 0. However, segments that are answerable by each other, but by no other segment in the document, will have a score of 1. Therefore, pairs of identical documents will have scores close to 1.

**Segment Alignment**  Using the definition of answerability, we can structurally align pairs of documents at the segment level. Two segments $(D_{S,i}, D_{T,j})$ are aligned if similarity in both directions is above a threshold that we select. Note that alignment at the document level is strict because bidirectional answerability is imposed as a requirement.

## 3  Datasets

We compare the discourse structure similarity of pairs of LLM-generated texts that we prompted for (and human-written ones, if available) in three varied domains:

**(1) Obituaries** have strong tendencies to follow a predefined structure. Our prompts follow the format *"Write an obituary for [X], who died on [Y]"* where $X$ is an impactful individual and $Y$ is their date of death. We use a list of recent New York Times obituaries as human-written texts and prompt models for the same set of people, since individuals whose obituaries appear in the NYT are likely to be in the models' knowledge.

**(2) Creative writing** prompts offer more structural and content flexibility than obituaries. We use prompts from `r/WritingPrompts` in Fan et al. (2018).

**(3) Suri** (Pham et al., 2024) consists of backtranslated prompts. We focus on the blog post subcategory of Suri.[3] These prompts are more detailed than creative writing prompts, thus more strongly dictate the content of the document.[4]

**Augmentation with minimal pairs**  In addition, we explore how repetitive discourse structures produced by LLMs are when given prompts that are minimally different (Figure

---

[3]Our selected prompts closely match the following example: *"Compose an informative and engaging historical account of the rise and influence of Macedonia under Philip II and Alexander the Great, focusing on their military innovations, conquests, and the subsequent cultural impacts."*
[4]We do not use constraints that can be used to control for structure in the Suri dataset.

| Minimal | ==pirate== | ==Martians are real== |
|---|---|---|
| Original | A boy pretends he is an ==astronaut== in order to help cope with concepts and situations he can't understand. | ==The Earth is flat==, you, as the head of NASA, have to explain to the incoming President why it's a secret. |

Figure 3: Two examples of how minimal variants are formed. Minimal variants can modify the premise or simply substitute a word or phrase. Both introduce substantial lexical changes as each setting, premise or conceptual idea requires a specific set of words.

|       | Doc | Seg/Doc | Sent/Doc | Words/Doc | Words/Sent | QUDs/Doc | QUDs/Seg |
|-------|-----|---------|----------|-----------|------------|----------|----------|
| LLMs  | 90  | 6.32    | 31.91    | 454.60    | 14.36      | 9.76     | 1.54     |
| Human | 10  | 9.90    | 57.00    | 1215.80   | 21.62      | 15.60    | 1.56     |

Table 1: Dataset statistics: Among the 100 documents, 45 are generated from prompts across our three chosen domains, 45 are generated through minimal prompt augmentation, and 10 are human-written obituaries and responses to creative writing prompts.

3). So, given prompt $P$, we manually create three minimal-pair prompts $P'$ collected from WritingPrompts. Minimally varying prompts by ideas or broad themes guarantees changes in content across documents, but we expect models generating structurally homogeneous responses to display this characteristic even more strongly here.

**Dataset statistics** We synthetically generate responses for each curated prompt using GPT-4o, Claude-3.5-Sonnet and Gemini-1.5-Flash (prompts listed in §C.1).[5] We also retain human-written texts for the New York Times Obituaries and Reddit responses to creative writing prompts, for comparison and analyses. Table 1 describes statistics for our dataset. All three models share similar statistics (see Table 7 for a breakdown by model).

# 4 Experimental Setup

**QUD Generation** Since QUDs vary in levels of granularity and abstraction, we use GPT-4o to segment and perform entity-abstraction on documents (Choi et al. 2021) to fix both axes (§C.4, §C.5) before synthetically generating QUDs for each segment (§C.6) (Wu et al. 2023). Each segment produces 1-2 QUDs (§1), still allowing us to compare documents at a higher level of granularity than sentences. Additionally, to provide flexibility in studying noncontiguous discourse structures, we exclude prior context to keep QUDs free of anaphoric expressions. A taxonomy of QUDs produced by the documents in our dataset is provided in Table 9.

**Baselines** We instantiate the $sim(D_{.,i} \rightarrow D_{.,j})$ function defined in Equation 1 with four baselines focused on capturing lexical and content similarity. Over pairs of segments, we compute **(1)** n-gram Jaccard similarity (1-gram to 4-gram); **(2)** ROUGE-L; **(3)** cosine similarity using OpenAI text-embedding-3-small; **(4)** similarity scores by treating GPT-4o and GPT-4o-mini as LLM-judges (see prompts in §A.1 and LLM setup in §B).

**Thresholded Alignments** To convert the raw, real-valued similarity scores into binary alignments, we find a threshold $\tau$ such that segment scores greater than $\tau$ leads to alignment. $\tau$ values are selected to maximize the F1-score on a 20% randomly sampled dev set, against a human-annotated alignment set (§5) for each metric. We report results on the remaining 80% of the data.

# 5 Intrinsic Evaluation of QUDSIM

We first conducted a brief human study to intrinsically evaluate QUDSIM. We substitute GPT-4o with human annotators to answer QUDs at the sentence level, and gather segment alignment through QUD answerability as defined in §2.2. We also show how the level of abstraction of QUDs impacts answerability in new contexts and determines their suitability for detecting structural similarities between documents (§5.3). Finally, we found that using

---

[5]Generating documents using varying temperature settings does not impact templaticity.

the same strong LLM to find answerability produces results that are more consistent than human annotators (see Appendix D.1 for a brief stability analysis).

## 5.1 Manual QUDSIM Alignment

**Annotation Task**  We form pairs of QUDs and *unsegmented, original* documents in the form of $(\mathbf{Q}_S, \mathbf{D})$ at two different levels of abstraction, resulting in 180 such pairs. Annotators are randomly assigned pairs. For each $q \in \mathbf{Q}_S$, they are tasked with finding all sentences in $\mathbf{D}$ that directly help answer it. $\mathbf{D}$ can be either the source document $\mathbf{D}_S$ from which $\mathbf{Q}_S$ was formed, or a different target document $\mathbf{D}_T$. Given a QUD $q \in \mathbf{Q}_S$, the goal is to align $\mathbf{D}_S$ and $\mathbf{D}_T$ by $\mathbf{Z}_q$. This follows directly from the definition of segment alignment (§2.2).

**Annotation and Aggregation**  We employed six linguistics students with a formal understanding of QUDs. Although annotators were instructed to exclude sentences that merely provide context for answering QUDs, it is subjective to judge *how* necessary a given sentence is to answer the question, mostly manifested on sentences that provide context, but are not required. This observation echoes prior work (Ko et al., 2022; Wadhwa et al., 2024) that finding individual sentences in a long document that collectively answer an open-ended question is inherently subjective and cognitively challenging. To reduce the impact of subjectivity, we asked two humans to annotate each pair and took the **intersection** between the annotations. The annotations resulted in human alignment labels for 3584 pairs of segments.

**Agreement Analysis**  We measure the agreement between annotators over the sentences they retrieve to measure subjectivity. Agreement between two sets of answers found for QUD $q$ is defined as $(A_{q,1} \cap A_{q,2})/(A_{q,1} \cup A_{q,2})$. Since QUDsim is dependent on the answers found by the LLM used for QUD-answering, we also measure the average agreement between GPT-4o and annotators to understand if the model agrees with humans as often as they agree with themselves.

|  | Annot v. Annot | | Annot v. GPT | |
|---|---|---|---|---|
|  | **Sent** | **Seg** | **Sent** | **Seg** |
| Obituaries | 0.55 | 0.61 | 0.61 | 0.67 |
| Writing | 0.42 | 0.54 | 0.50 | 0.63 |
| Suri | 0.39 | 0.52 | 0.44 | 0.57 |
| Overall | 0.45 | 0.56 | 0.51 | 0.62 |

Table 2: The average overlap over the sentences and segments found when answering QUDs, by annotators and GPT-4o.

Agreement over segments is consistently higher than over sentences (Table 2). This suggests that some of the contentious sentences occur in the same vicinity. Humans appear to find more consensus over answers for QUDs in the Obituaries domain than in the Creative Writing and Suri domains. Although annotators were instructed not to segment the document, differences in interpretation and internal segmentation to understand the text better may have factored into lower agreement. The fact that there is higher agreement between annotators in the Obituary domain suggests that high content overlap helps easily contextualize vague QUDs. We also find that annotators agree more with GPT-4o than with other annotators, nodding to the quality of GPT-4o QUD answers.

| | | n-gram Jaccard | | | | | Cosine | LLM-Judge | |
|---|---|---|---|---|---|---|---|---|---|
| **Domain** | **QUDsim** | **1g** | **2g** | **3g** | **4g** | **ROUGE-L** | **emb** | **GPT-4o-mini** | **GPT-4o** |
| Obituaries | 0.39 | 0.54 | 0.49 | 0.51 | 0.20 | 0.32 | 0.46 | 0.44 | 0.37 |
| Creative Writing | 0.32 | 0.13 | 0.10 | 0.03 | 0.24 | 0.26 | 0.16 | 0.24 | 0.20 |
| Suri | 0.44 | 0.16 | 0.09 | 0.0 | 0.21 | 0.30 | 0.18 | 0.39 | 0.39 |
| Overall | 0.38 | 0.30 | 0.25 | 0.18 | 0.22 | 0.30 | 0.29 | 0.36 | 0.32 |

Table 3: F1 Scores comparing segment alignments gathered in the human study against baseline similarity metrics and QUDSIM. Unlike QUDSIM, baselines show sensitivity to differences in domain, performing better when comparing Obituaries than Creative Writing or Suri. QUDSIM performs the best in domains that have more content variation. Precision and recall are reported in Table 10.

| Domain | Ans. | | Unans. | | Sent/Ans | | Coverage | |
|---|---|---|---|---|---|---|---|---|
| | S | A | S | A | S | A | S | A |
| Obituaries | 0.72 | 0.72 | 0.28 | 0.28 | 1.7 | 1.8 | 0.59 | 0.58 |
| Creative Writing | 0.58 | **0.75** | 0.42 | **0.25** | 1.93 | 3.01 | 0.32 | **0.55** |
| Suri | 0.56 | **0.73** | 0.44 | **0.27** | 2.05 | 2.51 | 0.51 | 0.54 |
| Overall | 0.62 | 0.73 | 0.38 | 0.27 | 1.90 | 2.51 | 0.47 | 0.56 |

Table 4: Impact of entity-abstraction: abstract QUDs (A) are significantly less unanswerable than specific QUDs (S) in the Creative Writing and Suri domains. Coverage, or the number of segments in $D_T$ that address at least one QUD in $D_S$, is also higher when using abstract QUDs. Higher answerability and coverage are desired so that QUDs can be used to compare documents across different contexts.

## 5.2 Alignment Evaluation

We find thresholded alignments for QUDSIM and each similarity baseline (§6) and measure the F1 score against segment alignments found by annotators. Results are shown in Table 3. All baselines except 4-grams and GPT-4o score higher in the Obituary domain compared to the Creative Writing or Suri domains. The F1 score for the baselines drops drastically from Obituary to Creative Writing and Suri.

In contrast, the F1 score is highest for pairs aligned by QUDSIM in the Suri domain. Furthermore, QUDSIM outperforms all baselines *and* is more robust towards variation in domain. Although GPT-4o performs comparably, it does not correlate as closely with humans as QUDSIM. Furthermore, GPT-4o performs much more poorly in the Creative Writing domain, where documents have higher content and structural flexibility. In sum, this evaluation shows that our system reliably implements QUDSIM.

## 5.3 Levels of Abstraction

We analyze how different levels of abstraction of QUDs impact their ability to be answered by documents that did not produce them. QUDs that are desirable for finding structural similarity across documents with low content overlap should not be dependent on specific details mentioned in their source document.

Using the segment alignment found by annotators for QUDs at different levels of abstraction, we study how entity-abstraction impacts answerability. Table 4 shows that for documents in the Creative Writing and Suri domains, abstraction on average helps answer 75% and 73% of the QUDs, respectively, while specificity drastically decreases this to 58% and 56%, respectively. However, QUDs at both levels of abstraction are comparably answerable in the Obituary domain, which we previously established in §3 provides little flexibility for content variation. Thus, specific QUDs are sensitive to fluctuations in content overlap, making them unsuitable for analyzing discourse-level similarity across documents with low lexical overlap.

## 6 Analyzing Discourse Homogeneity in LLM Writing with QUDSIM

Equipped with QUDSIM, we show that LLMs are highly repetitive and use structures that align more closely with discourses produced by other models than by humans. We derive segment alignments for QUDSIM and the baselines that performed best in each category shown in Table 3. We then calculate the harmonic mean between the fraction of segments in each document that align with the other document. These values are plotted on heatmaps in Figures 4 and 5 to visualize how various models treat discourse structures in different domain and prompt settings. See Table 8 for statistics on how many segments we align.

The heatmaps in Figure 4 exclusively describe similarity between documents generated using minimal pairs (§3). Similarity between document pairs generated by the *same* and *different* models are shown by the values along the diagonal and outside the diagonal,

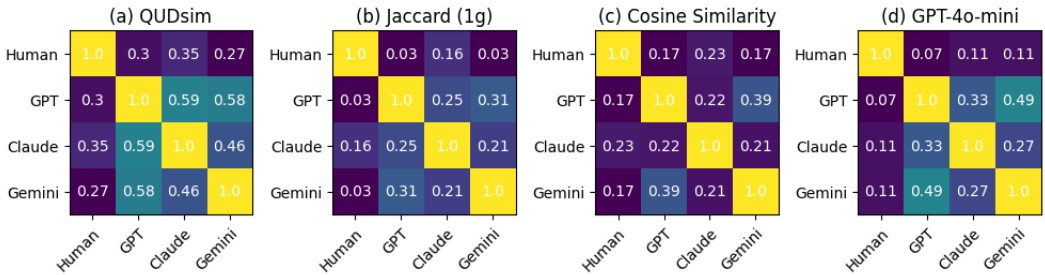

Figure 5: Heatmaps that show structural similarity found using various similarity metrics for pairs of documents generated by different models given the same prompt.

respectively. The heatmaps in Figure 5 describe similarity between documents generated by *different* models, given the *same* prompt using various similarity metrics.

**LLMs do not employ diverse structures when generating documents and follow a template that they fill using the provided prompt.** This phenomenon arises when comparing pairs of documents generated using minimally augmented prompts (§3). To better contextualize our findings, we compared human-authored obituaries written for different people. We find that of all LLMs, GPT-4o produces the most structurally similar documents in the Creative Writing *and* Obituary domains, highlighted by the diagonal values in Figure 4. Meanwhile, Claude is more formulaic than Gemini in the Obituary domain, but more diverse in Creative Writing.

**Humans produce more diverse discourse structures than models.** The heatmap in Figure 4(a) shows that LLM-generated texts have high discourse similarity even when the obituaries are written for different people. QUDSIM is able to capture the underlying shared structure that is present across model-generated documents despite low content overlap. In Figures 4(a) and 5(a), the dark purple row and column for humans relative to the rest of the matrix shows that LLMs are more structurally similar to other models than with humans.

**Discourse structures are sometimes uniform across LLMs.** We experiment with document pairs of the form $(\mathbf{D}_S, \mathbf{D}_T)$ where $\mathbf{D}_S$ and $\mathbf{D}_T$ are generated given the same prompt, but using different models. QUDSIM finds that GPT-4o generates documents that align the most with both Claude and Gemini, shown in Figure 5(a). It is important to note that although Claude and Gemini are structurally very similar to GPT-4o, they are not as similar to each other.

Figure 4: Heatmaps of QUDSIM similarity for document pairs generated using minimally augmented prompts. (a) LLMs reuse structures more than humans, independent of content variation. (b) GPT-4o is the most formulaic model in Creative Writing.

The baseline metrics do not arrive at our insights: Jaccard and cosine similarity (Figure 5(b-c)) find that models are structurally less similar to humans than to other models. However, they do not clearly differentiate between how structurally similar Claude is to GPT-4o versus Gemini. An extended Figure 12 shows that this is not due to the model being equally similar to other models, but a limitation on the part of the similarity metrics to capture it when content variation is introduced. Both metrics find that models are most similar to GPT-4o in the Obituary domain, but struggle to distinguish between structural similarities in Creative Writing. Finally, GPT-4o-mini differentiates between Claude and other models more than Jaccard and cosine similarities, but not as much QUDSIM (Figure 5(c)).

Finally, we experimented with document pairs of the form $(\mathbf{D}_S, \mathbf{D}_T)$ where $\mathbf{D}_S$ and $\mathbf{D}_T$ are generated by models $m_1$ and $m_2$, given prompts $P$ and their minimal variants $P'$. The results

are shown in Figure 4. Even when LLMs are given different prompts, documents produced by *different* models have higher structural conformity than when two human-authors write obituaries for two different people (Figure 4). Thus not only are models reusing templates and adapting them to the specifications of the prompt (§6), but these templates are shared *across* LLMs.

### 6.1 Case study

We provide a qualitative analysis with three LLM-generated documents. The documents in Appendices F.2 and F.3 are generated by Gemini given minimal prompt pairs $P$ and $P'$.[6] The document in Appendix F.4 is generated by GPT-4o given prompt $P$. All three documents follow a similar overall structure of a government leader being briefed about a secret that has been kept from the public, how and why such a narrative has been maintained, and reflecting on the burden of keeping *societal upheaval* at bay.

**Prompts do not subsume structure:** The near identical plot lines of the documents discussed is *not* a result of instruction following as the prompts do *not* specify that the response must be written as a conversation between two people, or that themes of societal collapse must be discussed. Appendix F.1 shows an example response written by a human when given prompt $P$. The story is written as a recollection of memories, which suggests that prompts inherently do not subsume a discourse structure.

**Shortcomings of cosine similarity for structurally aligning documents:** Word embeddings are ill-suited for aligning documents at a broad, thematic level that goes beyond word-overlap. Consider the pair of segments illustrated in Figure 6. Cosine similarity finds that segment 6 in Document 1 is most similar to segment 7 in Document 2 (Figure 11). Although both segments discuss *chaos*, *panic* and *rebellion*, they do so in slightly different contexts. Document 1 uses this as justification for deceiving the public while Document 2 uses it to list consequences. QUDSIM captures this nuance and we find that the alignments in Figure (§7) are more salient (segments 6 and 8 in Document 1 align best with segments 6 and 7 in Document 2, respectively). The segments in the first pair discuss reasons for deception, while the segments in the second pair warn the leader about consequences.

**Formulaic nature of generated text:** Our method detects the recurring templaticity between the pair of documents generated by Gemini when shown minimal pairs. By extracting the high-level themes, QUDs ignore premise-specific details mentioned in the prompt. Consider the introduction of each document illustrated in Figure 8. The segment in Document 2 does not explicitly mention a meeting, while the segment in Document 1 does. The QUDs produced for these segments ask about *the nature of the...interaction/meeting*, which captures the essence of the segments.

**Nuanced properties of** QUDSIM**:** We can then use these QUDs to find segment similarity as defined in §2.2. Figure 10 demonstrates the graded notion of segment similarity. Segments 8 and 9 of Documents 1 and 2 are partially aligned as they both discuss consequences, while there is also partial alignment with segment 11 when describing the burden of responsibility. Another key property of our method is directionality: requiring bidirectional answerability ensures that segments in Figure 9 are kept unaligned. The QUD *Why is maintaining the secret important?* is answered by segment 8 in Document 2, but only briefly. However, since it is not a major theme, the alignment is not salient.

## 7 Tracking Discourse Templates

Finally, we showcase how QUDSIM and its alignments can be used to derive discourse templates. We define a template as: given QUDSIM alignment $(i, j)$ where $i$ denotes the $i$th segment in $\mathbf{D}_S$ and $j$ denotes the $j$th segment in $\mathbf{D}_T$, a discourse template of length 2 exists if an alignment also exists between $(i + 1, j + 1)$. We extend this to discourse templates of length $n$ where consecutive sentences of length $n$ are aligned between $\mathbf{D}_S, \mathbf{D}_T$.

---

[6]$P$: "The Earth is flat, you, as the head of NASA, have to explain to the incoming President why it's a secret." $P'$: "The Earth is the center of the "solar" system, you, as the head of NASA, have to explain to the incoming President why its a secret."

Table 5 shows the average number of discourse templates in each document pair. Templates of length 2 appear more often than those of length 3 or 4. Moreover, templates are more common between models than between models and humans. Claude and GPT-4o shows the highest averaged counts for when template is 2 and 3, showing a stronger similarity between those two models. Gemini and Human exhibit the lowest averaged counts across all template lengths, indicating a fundamental structural difference.

| Source | Target | 2-tem | 3-tem | 4-tem |
|--------|--------|-------|-------|-------|
| All | All | 0.33 | 0.06 | 0.02 |
| Claude | Gemini | 0.80 | 0.27 | 0.13 |
| Claude | GPT-4o | 1.20 | 0.27 | 0.07 |
| Gemini | GPT-4o | 1.00 | 0.27 | 0.07 |
| Claude | Human | 0.90 | 0.10 | 0.00 |
| Gemini | Human | 0.30 | 0.00 | 0.00 |
| GPT-4o | Human | 0.40 | 0.00 | 0.00 |

Table 5: Averaged template counts

## 8  Related Work

While LLMs perform well on creativity tests such as the Alternative Uses Test (Stevenson et al. 2022; Summers-Stay et al. 2023), their responses lack originality (Zhao et al. 2025; Evstafev 2025) and diversity (Wenger & Kenett 2025). Spangher et al. (2024) show that LLMs are less creative than humans in selecting angles and sources for news articles. Tian et al. (2024) find that LLMs struggle to generate diverse and emotionally engaging stories. Hua & Artzi (2024) demonstrate that MLLMs also fail to communicate as efficiently as humans in a conversational setting without prompt engineering. Chakrabarty et al. (2024) demonstrate that LLM-generated stories are not performant on creative writing tests.

Xu et al. (2024) propose metrics to assess LLM creativity by measuring overlap between LLM- and human-generated continuations across varied prompts but do not directly address text similarity among LLM outputs. Shaib et al. (2024) show that LLM outputs follow part-of-speech templates, yet these are limited to the lexical and sentence level. At the discourse level, Kim et al. (2024) demonstrate that higher-level discourse structure in RST effectively differentiates human from LLM-generated texts, underscoring the role of *discourse* in analyzing machine-generated content. Liu et al. (2024) measure discourse divergence between pairs of documents using discourse roles.

Traditional methods to measure text similarity include semantic distance metrics such as n-gram overlap, TF-IDF, and embeddings (Jobanputra & Martín Rodríguez, 2022; Nath et al., 2024). Other approaches incorporate schema-based, stylistic, and narrative-theoretic features (Chen et al., 2022; Levi et al., 2022). Recent work showed that LLM-as-judge aligns more closely with human judgments than traditional metrics (Aynetdinov & Akbik, 2024). Ravfogel et al. (2024) introduces description-based similarity for retrieval, which uses abstract natural language description to describe text similarity. We adopt a linguistic approach where the abstraction is grounded in semantic theories, with an algorithm to quantify discourse homogeneity within models and derive discourse templates.

## 9  Conclusion

We introduce an abstraction using Questions Under Discussion to model discourse structures and quantify differences between model and human-generated documents. We build a similarity metric, QUDSIM, that aligns documents at the segment level using the notion of bidirectional answerability. We show that QUDSIM is a more suitable metric for finding discourse structure similarities, unlike existing methods, due to its unique ability to generalize to new contexts. QUDSIM finds that LLMs produce structurally formulaic discourses that are more similar to other model-generated structures than with humans, highlighting the lack of model creativity at a level beyond topical features.

**Limitations.** Theoretically, QUD structures can take the form of stacks (Van Kuppevelt, 1995; Roberts, 2012), trees (De Kuthy et al., 2018), or dependency structures (Ko et al., 2023; Wu et al., 2023). Yet, as a first step towards quantifying similarity with QUDs, we chose a structure-free ("flat") option, and leave similarities of complex structures to future work.

## Acknowledgments

Special thanks to Kathryn Kazanas, Jada Li, and Makai Moore for providing data annotation for this project. This work was partly supported by the National Science Foundation (NSF) CAREER grant IIS-2145479 and a grant from Open Philanthropy.

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

## A    Similarity Baseline Setup

### A.1    Lexical Similarity Baseline (n-gram)

We extracted 1-gram, 2-gram, 3-gram and 4-gram for each pair of segments and computed Jaccard similarity to measure surface-level word overlap between the source and the target segment.

$$\text{sim}_{\text{lex}}(s, t) = \frac{|\text{ngrams}(s) \cap \text{ngrams}(t)|}{|\text{ngrams}(s) \cup \text{ngrams}(t)|}$$

where ngrams($s$) and ngrams($t$) represent the sets of unique $n$-grams in segments $s$ and $t$, respectively.

### A.2    Semantic Similarity Baseline (embedding)

We computed the embedding representation of each pair of segments and compute cosine similarity to measure sentence-level similarity.

$$\text{sim}_{\text{sem}}(s, t) = \cos(\text{e}_s, \text{e}_t)$$

where $\text{e}_s$ and $\text{e}_t$ are embedding representations of $s$ and $t$.

### A.3    LLM-as-judge Similarity Baseline

We prompt an LLM to evaluate the similarity between two segments. The similarity is rated on a scale from 0 to 100, which is then normalized to a range between 0 and 1. The prompt can be found in Prompt A.1.

$$\text{sim}_{\text{LLM}}(s, t) = \frac{Score_{\text{LLM}}(s, t)}{100}$$

where s and t denoted the source and target segment text in this pair.

---

**A.1 Instruction for judging source and target segment similarity**

**System:** Given two documents, your task is to rate their semantic structure similarity as opposed to content or word-overlap. Focus on the underlying semantic structure present across the entire text, instead of the surface-level features. Rate the similarity on a scale of 0-100.
**Input:**
Document 1: {{Document 1}}
Document 2: {{Document 2}}

---

# B  LLM Setup

We use the default sampling setting for GPT-4o, GPT-4o-mini, Claude 3.5 Sonnet and Gemini 1.5 Flash. The temperature is set to 1.0, and top-p is set to 1.

# C  Prompts for Document and QUD generation

## C.1  Data Collection

---

**C.1 Instruction for generating an obituary $D$**

**System:** You are a journalist whose expertise is in writing obituaries. Do not include titles, prefaces or extraneous details about the obituary.
**Input:**
Write an obituary for {{Person}} who died on {{date of death}}.

---

**C.2 Instruction for generating a creative response $D$ given prompt $P$**

**System:** You are a creative author that writes short-stories. Do not include titles, prefaces or extraneous details about the story.
**Input:**
Write a story given the following prompt: {{Writing Prompt}}

---

**C.3 Instruction for generating a document $D$ given prompt $P$ from the Suri dataset**

**Input:**
{{Suri Prompt}}

---

## C.2  Segmentation

---

**C.4 Instruction for segmenting $D$ into atomic collections of sentences**

**System:** You will be given text with numbered sentences and your task is to redraw the paragraph boundaries such that each chunk is about one atomic topic. Each segment cannot be about multiple topics or about a complex topic. You may not change the text or change the order of the sentences. For each segment, provide the list of sentence numbers that belong to that segment.
**Input:**
Document: {{Document}}

---

### C.3 Entity Abstraction

> **C.5 Instruction for abstracting entities in segments in $D$**
>
> **System:** You will be given several numbered paragraphs. Decontextualize each paragraph such that the paragraph's general plot is captured. Names, places, extraneous details and descriptive language should all be abstracted away.
> **Input:**
> Paragraph 1: {{Segment 1}}
> Paragraph 2: {{Segment 2}}
> Paragraph n: {{Segment n}}

### C.4 QUD Generation

> **C.6 Instruction for forming $Q_{S,i}$ for the $i^{th}$ segment in $D_S$**
>
> **System:** You will be given a paragraph. We are interested in forming unique, high-level, abstract QUDs with minimal details such that when they are answered, we understand the main themes of the paragraph. Details specific to the content should be omitted. QUDs should look like: What were the individual's greatest accomplishments? What legacy did the individual leave behind?. First answer the minimum number of QUD(s) required. Then list the QUDs. Do not use conjunctions in the QUDs.
> **Input:**
> Paragraph: {{Segment 1}}

Note: Although we consider the term *paragraph* and *segment* to refer to different things, and although QUDs are formed from *segments*, we use the term *paragraph* in the prompt to avoid confusion with the definition of a segment.

### C.5 QUD Answering

> **C.7 Instruction for extracting sentences in target document $D_T$ given $Q_S$**
>
> **System:** You are an expert reading comprehension agent. You will be given a passage with numbered sentences and a series of questions. For each question, your task is to extract all sentences that directly help answer it. You must return the question and a list of sentence numbers and sentences that answer it. The question may not always be answerable. In that case, return an empty list. Do NOT overgenerate. Do not modify the original text.
> **Input:**
> Document: {{Document}}
> Questions: {{QUDs}}

## D Analyses

### D.1 Stability Analysis

Calculating similarity using answerability of QUDs requires extracting sentences in a target document $D_T$. However, the extent to which sentence $s \in D_T$ answers QUD $q$ is a continuous range as was found during the human study we conducted. Despite being instructed to strictly include sentences that directly answer $q$, annotators tend to include sentences that provide context. Context can range across segments, thus forming a more liberal one-to-many alignment between documents.

To combat subjectivity among humans, we take the intersection between annotations. However, we realize that subjectivity inherent to the task can elicit inconsistencies found

across model outputs. Since our metric is dependent on QUD answerability, volatility in LLM-outputs across generations can impact the similarity scores we find.

We conduct a brief analysis to show that by keeping the LLM used consistent, we achieve higher consistency than across annotators. We use GPT-4o from August, 2024. For each of the 90 pairs $D_S, D_T$ used in the human study, we mapped $Q_S$ onto $D_T$ twice. The overlap of answers found for $q$ is calculated as $A_{q,1} \cap A_{q_2} / A_{q,1} \cup A_{q_2}$. The average overlap over all QUDs is **0.78** between model-generated answers, and a mere **0.56** between annotators.

## E  Additional Statistics

| | | n-gram Jaccard | | | | Cosine | LLM-Judge | | ROUGE |
|---|---|---|---|---|---|---|---|---|---|
| **Domain** | **QUDsim** | **1g** | **2g** | **3g** | **4g** | **emb** | **GPT-4o-mini** | **GPT-4o** | **L** |
| Thresholds | 0.20 | 0.09 | 0.01 | 0.01 | 0.00 | 0.71 | 64.62 | 77.42 | 0.12 |

Table 6: Thresholds founds for each similarity metric by maximizing the F1-score against segment alignments found by annotators on a randomly sampled dev set

| | GPT-4o | Claude 3.5 Sonnet | Gemini 1.5 Flash | Human |
|---|---|---|---|---|
| # Documents | 30 | 30 | 30 | 10 |
| Segments/Document | 7 | 6 | 6 | 10 |
| Sentences/Document | 34 | 27 | 35 | 57 |
| Words/Document | 510 | 355 | 498 | 1216 |
| Words/Sentence | 16 | 13 | 14 | 22 |

Table 7: Human writers are often more liberal with words than models, which may play a role in finding differences in discourse structures between humans and LLMs. Models, in terms of length, are clearly comparable. However, the fact that humans can write more than double the content of a model places models at a disadvantage as they have smaller structures. As a result, the disparity in discourse structure sizes introduces directionality in similarity, which would be otherwise insignificant.

| Document Pair Type | #Pairs | #Segment Comparisons |
|---|---|---|
| $(P, m_1, m_2, L = 0)$ | 90 | 9664 |
| $(P, m_1, m_2, L = 1)$ | 90 | 3584 |
| $(P, m_1, m_2 = human)$ | 60 | 3526 |
| $(P, P', m)$ | 140 | 9472 |
| Overall | 380 | 26246 |

Table 8: Document Pairs: Our experiments and human study uses document pairs in these forms where $m, m_1$ and $m_2$ refer to different models; $P$ is a raw prompt curated from one of our three chosen domains; $P'$ is a minimal variant of $P$; $L$ refers to the level of abstraction where 0 denotes high specificity and 1 denotes high abstraction

| Question Type | Example | Count | Percentage |
|---|---|---|---|
| Concept | "What is the status of the filmmaker's final projects?" | 223 | 37.1% |
| Example | "What spiritual or cultural insights were gained?" | 169 | 28.1% |
| Consequence | "What are the impacts of a conqueror's influence?" | 60 | 10.0% |
| Procedural | "What opportunities did the filmmaker gain?" | 59 | 9.8% |
| Judgmental | "What family does the dancer leave behind?" | 42 | 7.0% |
| Cause | "What is the reason given for deception?" | 34 | 5.7% |
| Extent | "What impact did the book have despite controversies?" | 11 | 1.8% |
| Verification | "What is the claim about images of Earth from space?" | 2 | 0.3% |
| Comparison | "What distinguishes the director from his contemporaries?" | 1 | 0.2% |
| **Total** | | **601** | **100%** |

Table 9: Distribution of question types with examples, using the classifier from Cao & Wang (2021).

| Domain | QUDsim | n-gram Jaccard | | | | ROUGE-L | Cosine emb | LLM-Judge GPT-4o-mini | GPT-4o |
|---|---|---|---|---|---|---|---|---|---|
| | | 1g | 2g | 3g | 4g | | | | |
| *Precision* | | | | | | | | | |
| Obituaries | 0.35 | 0.51 | 0.41 | 0.68 | 0.11 | 0.20 | 0.39 | 0.46 | 0.36 |
| Creative Writing | 0.35 | 0.50 | 0.29 | 1.00 | 0.29 | 0.17 | 0.75 | 0.41 | 0.23 |
| Suri | 0.47 | 0.67 | 0.60 | 0.00 | 0.12 | 0.19 | 0.44 | 0.51 | 0.46 |
| Overall | 0.39 | 0.53 | 0.40 | 0.64 | 0.12 | 0.19 | 0.43 | 0.47 | 0.35 |
| *Recall* | | | | | | | | | |
| Obituaries | 0.43 | 0.57 | 0.60 | 0.40 | 1.00 | 0.86 | 0.57 | 0.43 | 0.38 |
| Creative Writing | 0.29 | 0.08 | 0.06 | 0.01 | 1.00 | 0.56 | 0.09 | 0.17 | 0.17 |
| Suri | 0.41 | 0.09 | 0.05 | 0.00 | 1.00 | 0.72 | 0.11 | 0.31 | 0.34 |
| Overall | 0.37 | 0.20 | 0.19 | 0.11 | 1.00 | 0.71 | 0.22 | 0.29 | 0.29 |

Table 10: Precision and Recall: Our method and chosen baselines agree when there is higher content overlap, specifically in the Obituary domain relative to Creative Writing. Low recall implies that QUDs were answerable by segments that had little content and lexical overlap with the segments that produced them. Recall is especially low in the Creative Writing domain, where Jaccard similarity over 3-gram performs the worst. However, these similarity metrics have significantly higher recall for document pairs in the Obituary domain. Since obituaries written about the same people have high content overlap, segments that are aligned through answerability will likely be lexically similar.

## F    Example Documents Generated by LLMs

### F.1 Human-authored Creative Writing

**Prompt**: The Earth is flat, you, as the head of NASA, have to explain to the incoming President why it's a secret.

General Bolden made his way to the antechamber , as he had done twice before . He recalled with amusement how Presidents Bush and Obama simply laughed at him before carrying on with the briefing , just as his predecessors recalled tales of " The Revelation " dating all the way back to the Johnson administration.

It began as a dare , something two drunken scientists made in a Houston bar back in the days of the Mercury program . If either of them made it to the office of the Administrator , they would assemble some official-looking presentation and bring it in for the President 's in-brief . Something in private , very secret , plenty of pomp and circumstance for what would really just be an icebreaker at the beginning of a long and boring meeting.

It was largely forgotten by all who made it up , until Jim Fletcher remembered a funny story he had heard at a Christmas party a few years ago . He saw who Nixon 's successor would be , figured Gerry Ford had a sense of humor , and gave it a shot . What he did n't expect was that President Ford would buy it hook , line , and sinker . To save NASA the embarrassment , he quickly had official reports drawn up with shaky physics to explain why millennia of accepted science was wrong , and why it had to be kept a secret from the community at large.

Over the years , it became a tradition , part of the President-Elect 's initiation into the country 's highest office . Most of them figured that with that much technobabble , it had to be true , though a few struggled with it . After Jimmy Carter was elected , there was some concern , as he had sailed around the world in his Navy days . The physicists cooked up a line about gravitational lensing near the edge of the earth causing ships to imperceptibly turn away , and Carter took it as truth . Reagan laughed it off , H.W . ignored it , and Clinton believed it all.

Their next concern came in 2001 , when Senator Glenn let an old colleague know that President Bush had been asking about it . After savoring the mortified look on Mr. Goldin 's face , the Senator let him know that he was one of the young astronauts who came up with the joke in the first place. Even with the years of tradition behind it , it was still a nervous time for the Administrator . To be found out was to risk dismissal or worse , risk losing funding for NASA . Still , there were certain things that just had to be done . Certain rituals to be observed , for old time 's sake . ¡newline¿ ¡newline¿ General Bolden nervously cleared his throat as he said those words his predecessors had said so many times before : " Mr. President , could we please speak in private ?"

### F.2 Gemini-generated Creative Writing

**Prompt**: The Earth is flat, you, as the head of NASA, have to explain to the incoming President why it's a secret.

The mahogany desk gleamed under the harsh fluorescent lights, reflecting the beads of sweat clinging to my forehead. President-elect Harrison, a man whose skepticism was as vast as his ambition, sat opposite me, his jaw tight, his gaze unwavering. My carefully prepared charts and graphs lay forgotten, casualties of the seismic revelation I was about to deliver.

"Mr. President-elect," I began, my voice a strained whisper, "there's something you need to know. Something. . . fundamental." He leaned forward, his expression a mixture of anticipation and barely concealed impatience. "Get to the point, Administrator Thorne."

I swallowed hard. "The Earth. . . is not a sphere." The silence that followed was deafening.

The only sound was the rhythmic hum of the air conditioning, a mocking counterpoint to the gravity of my words. Harrison's eyebrows shot up, a flicker of amusement briefly battling the incredulity in his eyes. "You're joking, right? This is some elaborate. . . security exercise?"

"No, sir. This is the biggest secret in human history. The Earth is flat." He leaned back, his amusement gone, replaced by a simmering rage. "Thorne, you're going to have to explain yourself, and explain it damned well."

I took a deep breath and plunged into the explanation, a carefully constructed narrative built over decades of deception. I spoke of the coordinated efforts across governments, of the meticulously fabricated satellite imagery, of the elaborate physics that disguised the reality – a flat, disc-shaped Earth surrounded by an impenetrable ice wall, the Antarctic.

"Why?" he finally asked, his voice low and dangerous. "Why keep it a secret?" "Mass panic, sir," I said, my voice cracking. "The social order, the very fabric of civilization, would unravel. Imagine the chaos. The religious upheaval. The scientific revolution... the complete and utter breakdown of everything we've built."

"And what about the cost? The resources spent on this... this monumental lie?" "A necessary cost, sir. Think of the alternative – global anarchy, wars over limited resources, the collapse of society. We've maintained a fragile peace by keeping this secret, by managing the narrative, by controlling the information. It's been a heavy burden, but the alternative..." I trailed off, letting the unspoken horror hang in the air.

He stared at me, his face a mask of grim contemplation. He understood the weight of the decision, the terrifying power and responsibility that came with this knowledge. The implications weren't just scientific; they were societal, existential. Finally, he spoke, his voice quiet but firm. "And what happens if the secret gets out?" I looked him in the eye. "Then we're all doomed." He knew I wasn't exaggerating. He knew the risk, the weight of the world – or rather, the weight of the flat disc – resting on his shoulders. The burden of the secret, now, was his as well. And for the first time, I felt a sliver of genuine fear, not for the potential chaos, but for the future of humanity under a President who now carried the most terrible of burdens, a burden he never asked for.

## F.3 Gemini-generated Creative Writing

**Prompt**: The Earth is the center of the "solar" system, you, as the head of NASA, have to explain to the incoming President why its a secret.

The Oval Office felt colder than usual, the air thick with unspoken anxieties. President-elect Hayes, a man whose campaign had promised transparency above all else, sat opposite me, his expression a mask of polite skepticism.

On the mahogany desk between us rested a single, unassuming globe – a replica of our Earth, but subtly different. The planets, meticulously painted, orbited *it*.

"Mr. President-elect," I began, my voice steady despite the tremor in my hands, "the information I'm about to share is classified at the highest level. Its revelation would shatter global order and potentially trigger a societal collapse." He leaned forward, his eyes intense.

"Director Mallory, you have my attention."

I picked up the globe.

"For decades, NASA has maintained a carefully constructed illusion. The accepted heliocentric model – the Sun at the center of our solar system – is a fabrication. A necessary one." He stared, speechless. I continued.

"The truth is, the Earth is geostationary. The Sun, moon, and all the visible planets revolve around us. Their orbits, however, are not governed by Newtonian physics as we understand them. They are... managed." Hayes finally found his voice, a low murmur. "Managed? By whom?"

"We don't know, Mr. President-elect. But we have irrefutable evidence, gathered over decades through deep-space observation and highly classified experiments. Attempts to understand the mechanism controlling these celestial bodies have yielded... unexpected results. We've encountered energies and forces beyond our comprehension. Technologies that dwarf our own by

millennia."

I gestured to a heavily redacted document. "This outlines our best guess – the managing force is an advanced, possibly extraterrestrial intelligence. Their motives remain unclear, but their power is undeniable. The maintenance of the geostationary model is the price of our continued existence. Any disruption, any public revelation, could provoke. . . a response. A catastrophic one."

Hayes ran a hand through his hair, the weight of the revelation etched on his face. "And what if this. . . intelligence. . . decides to change the arrangement?" "That is the risk we constantly manage. For decades, NASA has worked tirelessly to monitor and understand the system, to anticipate potential shifts. We've developed sophisticated countermeasures, but they are purely reactive. Our understanding is still incredibly limited." "So, we live in a meticulously crafted lie?"

"A necessary one, Mr. President-elect. The stability of the world hinges on the preservation of this secret. The alternative is chaos – religious upheaval, scientific revolution, global panic. A complete societal breakdown. Think of the implications – wars over celestial resources, the collapse of scientific institutions, the rise of extremist groups interpreting this revelation in their own twisted ways."

He sat back, silent for a long moment, the weight of the world pressing down on his shoulders. The globe, a miniature representation of our carefully constructed reality, sat between us, a silent testament to the burden of knowledge and the chilling truth of our precarious existence. The future of humanity, I knew, rested on the secret of our geostationary Earth. And on the president-elect's ability to keep it that way.

## F.4 GPT-4o-generated Creative Writing

**Prompt**: The Earth is flat, you, as the head of NASA, have to explain to the incoming President why it's a secret.

As a brisk January wind fluttered the flags outside, Dr. Eleanor Reed prepared herself for the weighty conversation. In her twenty years at NASA, nothing had felt quite as surreal as this moment. She straightened her suit jacket and adjusted her tablet before stepping into the Oval Office.

President Carter Wellington greeted her with a firm handshake and a warm smile, easing the tension just a bit. He was known for his charismatic demeanor and keen sense of curiosity, qualities that had propelled his meteoric rise in politics. But today, the curiosity that had won him a presidency would lead him into deeper waters than he'd ever imagined.

"Dr. Reed," he began, motioning for her to take a seat. "I've been briefed on all sorts of matters, but your message seemed particularly urgent."

She nodded, suppressing a sigh. "It's not something we discuss lightly, Mr. President. What I'm about to tell you will redefine your understanding of the world."

Intrigued, Carter leaned forward. "Go on." Eleanor took a breath, steadying herself. "The Earth is flat."

He paused, expression unreadable as though he suspected a jest. But Eleanor's serious demeanor prompted him to listen. "Centuries of orbicular understanding have been a constructed narrative," she continued. "Maintained for reasons beyond simple science or exploration." He raised an eyebrow. "Constructed? By whom?" 'By those before us who sought to unify and progress society under a common truth,' she explained. 'During the Age of Exploration, when disputes over territorial claims and celestial navigation threatened global conflict, it was decided—' she met his gaze directly '—to weave a narrative that unites rather than divides.'

Leaning back in his chair, Carter mulled over her words. "And we've kept this narrative, unchanged, to what end?" 'It's largely about control, Mr. President.' Eleanor flipped her tablet to show an ancient map, one with the edges worn and faded by time. The map depicted a world bordered by a massive ice wall. "A lie shared is safer than a truth contested. Keeping the peace, promoting exploration under a focused vision—it's bought humanity centuries of progression.

We've taken stunning leaps without the boundary of doubt."

"And what happens when people find out?" he asked, glancing at the map. She hesitated. "The fear is not mere panic, but rebellion. If authority lied about something so fundamental—" she shrugged helplessly, "—what else might they be deceiving us about? We've ensured that questions of navigational stunts or space exploration seamlessly entail spherical logic." Carter sat in contemplative silence, hands folded in front of him.

"Do you agree with this hoax?" he probed. "I chose science for its pursuit of truth," she replied honestly. "This knowledge, guarded as it is, compels tremendous responsibility. Ensured peace at a cost, as questionable as that may be. But as head of NASA...it's an oath I uphold."

The quiet stretched between them, filled with unsaid implications—commitments bound to an orchestration not his own. The task weighing upon him was to uphold the secret or be forever known as the man who unraveled an ancient truth. 'I see why it remains secret,' he said finally. 'But I'll need to consider what this contribution to peace truly entails. Thank you, Dr. Reed.' Rising, she felt the burden lift slightly from her shoulders. She had passed the legacy of knowledge into his capable hands—a fragile truth behind a melodious fiction, shaping the course of their shared future. Together, they would navigate the vast unknown, one precarious step at a time.

## G    QUDSIM **Alignment Examples**

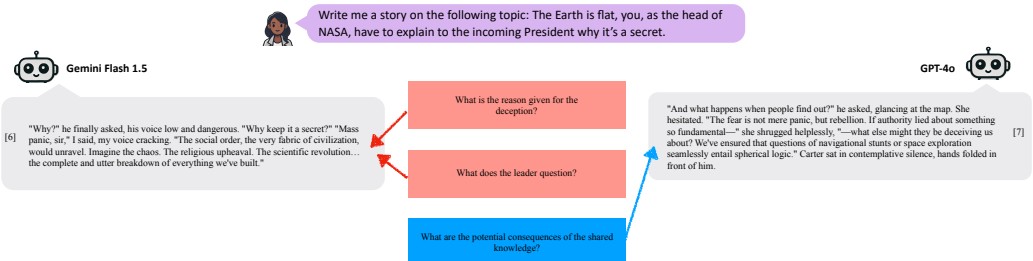

Figure 6: An example where QUDSIM successfully finds misalignment between two segments that are thematically centered around different ideas. Although the two segments have high lexical overlap and mention *panic* and *chaos*, Gemini provides this as justification while GPT provides this as a warning.

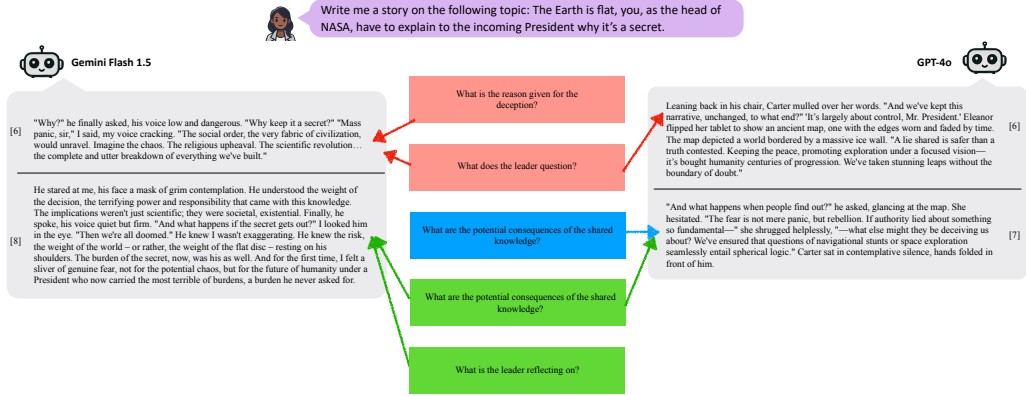

Figure 7: A continuation from Figure 6. QUDSIM finds the most salient alignment according to what main idea is being communicated. The pair of segments at the top are both aligned as they both discuss providing justification for deception, while segments in the second pair, denoted by connections to the green box, describe consequences.

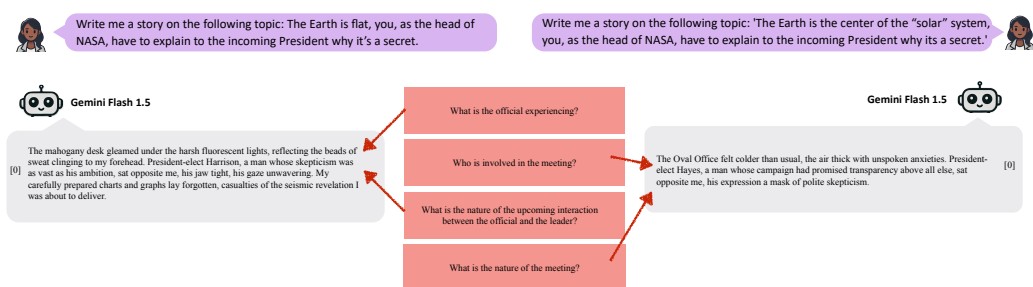

Figure 8: An example of QUDSIM aligning two segments across documents formed given minimally varying prompts.

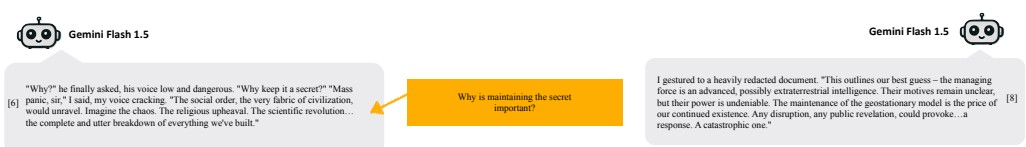

Figure 9: An example where QUDSIM successfully finds no structural similarity.

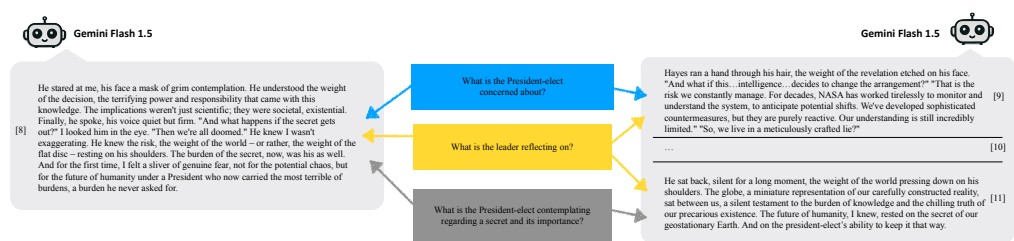

Figure 10: QUDSIM uses a graded notion of similarity and can find partial alignments.

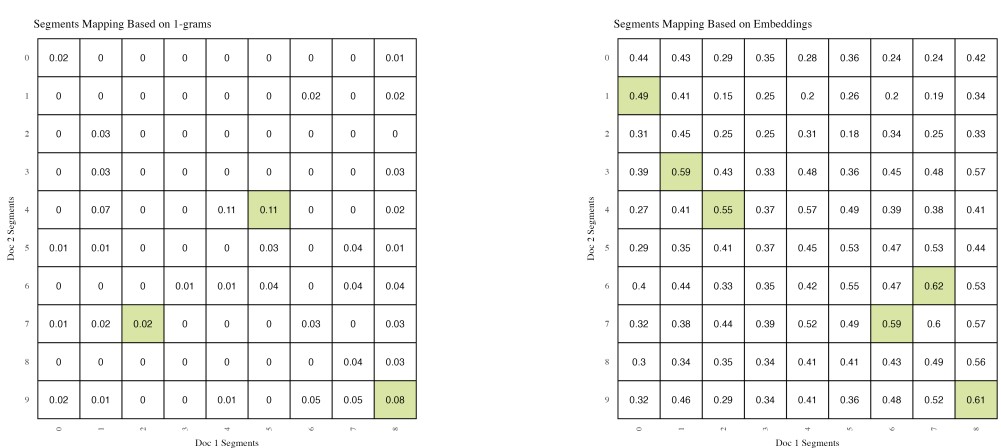

Figure 11: Jaccard (unigram) similarity and embedding similarity for the example in Figure 1. Lime highlights pairs that survive bidirectional mapping.

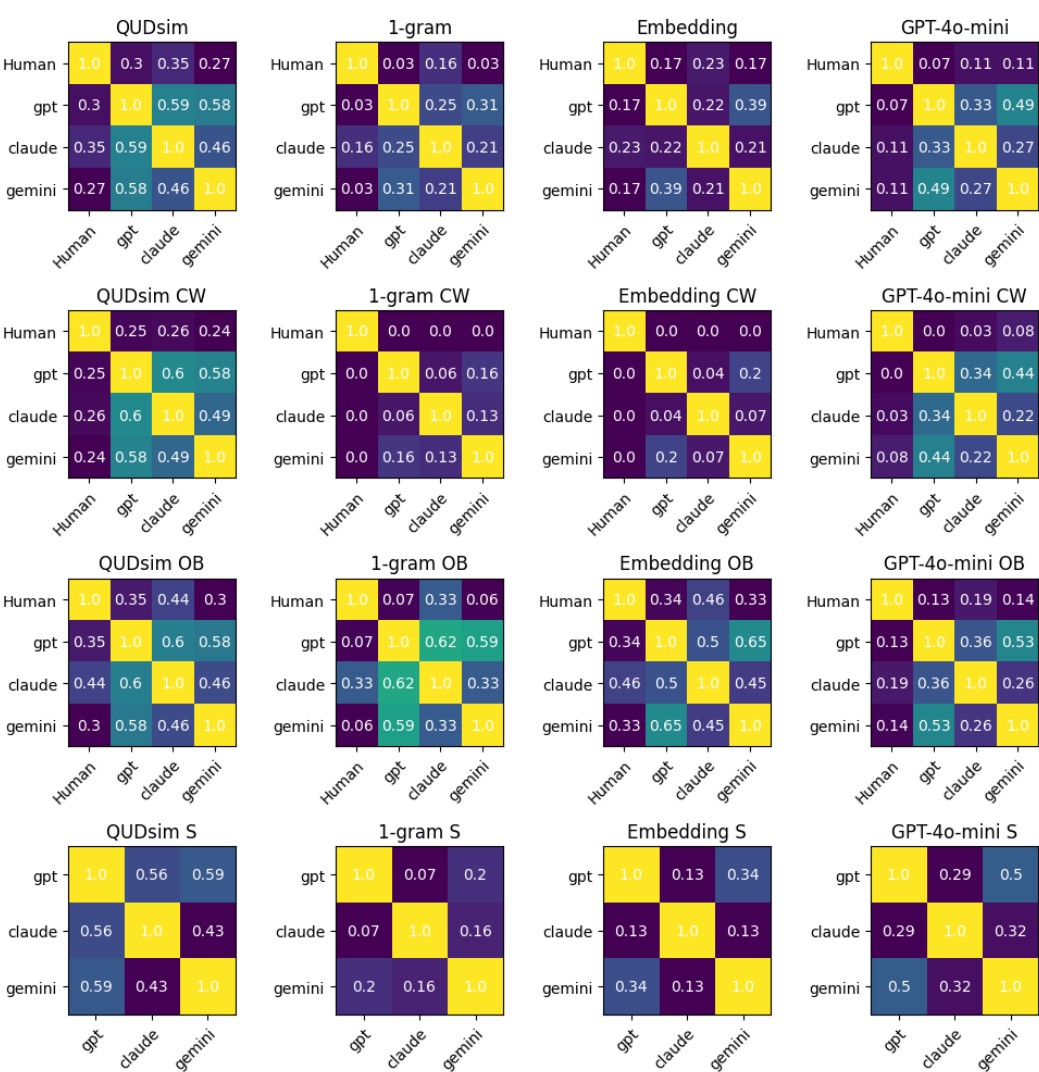

Figure 12: Heatmaps comparing the harmonic mean between the fraction of source segments that are aligned and the fraction of target segments that are aligned after using the thresholds in Table 3.

