# OpenReview forum: "QUDsim: Quantifying Discourse Similarities in LLM-Generated Text"
_colmweb.org/COLM/2025/Conference — COLM 2025_

### Official Review · Reviewer_UMuD · 2025-05-10

**Rating:** 8
**Confidence:** 4
**Ethics Flag:** 1

**Summary:**

The paper reports on a framework based on i) questions under discussion (QUD) and ii) alternative semantics to evaluate structural similarities in LLM-generated text. The main idea is that these similarities are not only limited to sentence or semantic patterns, but also to discourse-level features, and that in fact discourse-level analysis gives you more insights about these similarities.

The method is completely LLM-reliant. LLMs are used to generate texts, LLMs are used to segment the texts and then to generate the questions to analyze them. Comparisons with human data are provided though. LLM-generated texts are compared with human data, and LLM-segments are also compared with human-segmented data.

Solid evaluations of text coherence are hard and this is a good approach to that. There's a tension between content and structure when it comes to the evaluation of coherence and this framework strikes a good balance in terms of the comparison and shows that LLM-texts have more similarities with other LLM-texts than with human texts (that being the main result).

**Questions To Authors:**

- I was surprised to see that the human texts are longer than the LLM-generated ones (table 1), since LLMs tend to be very verbose. Could the genre have an impact on length (table 1 is not divided into genres)?

- how much would you expect generated-texts to approach the human-generated texts by refining the prompt?

- about the agreement of humans with chatGPT over other humans in the segmentation task, could it be that there may be possible segmentations possible and that humans are just more creative in their interpretations?

- in section 4, lines 146-151: I'd appreciate some clarification on the statement regarding the exclusion of prior content to keep QUDs free of anaphoric expressions. Could you elaborate on what this entails? Specifically, are the generated texts being modified in any way? I'd think that limiting anaphoric expressions could lead to more formulaic texts, which may diverge from the patterns typically observed in the human-generated texts.

Small stuff:

- line 56: what do you mean by "this aspect"?

- I was very confused as to how the questions were generated. Then the explanation came in page 5, section 4. Maybe you could add a sentence or two right in the introduction clarifying this.

- the text in figure 1 is extremely small when you read the paper in printed form

**Reasons To Accept:**

- the method is grounded in linguistic theory

- the method is label-free which is not a given since discourse tends to be very oriented towards structure, i.e, discourse parsing

- the paper contributes to research on differentiating machine-generated text from human-generated text, something very relevant nowadays

**Reasons To Reject:**

- technical details that may have an impact on the result are missing: decoding strategy used, temperature, etc... these are needed to ensure somewhat consistent generation results. (I assume they were the default ones for the experiments presented).

- the generated texts depend heavily on the prompt and because prompts can be endlessly refined to produce exactly what you want, it raises the question of whether the underlying premise truly holds

---

> ### Author Response · Authors · 2025-06-02
>
> Thank you for your feedback! Addressing the concerns raised below:
>
> > **technical details that may have an impact on the result are missing**
>
> We use the default parameters (temperature=1 and top-p=1). We included these details in Appendix B.
>
> > **the generated texts depend heavily on the prompt and because prompts can be endlessly refined to produce exactly what you want**
>
> It is true that prompting can heavily impact the results, but we believe the question we are studying is natural for two reasons. First, relatively short prompts are often used in practice: for instance, the WritingPrompts dataset from reddit features short instructions, and human writers working on an initial draft may not have a large amount of content worked out.
>
> Second, there is a fundamental question of how much of the space of human-written stories is being captured by LLMs. A short prompt represents an *easy* form of this task, as there are many possible answers, yet we see LLMs generating templatic ones. As a result, we see our findings as meaningful and still impactful even in the presence of longer prompts (perhaps until the prompt gets so detailed than LLM is just minorly revising an existing text, at which point we do not expect much deviation from the source).
>
> > **human texts are longer than LLM-generated ones**
>
> LLMs have a “natural” length of responses they tend to generate, which may be long for simple queries but may fall short in longer-form generation settings, particularly something like an obituary which could cover a wide range of facts about a person. Furthermore, there would be differences in terms of words per sentence depending on the genre. Creative writing includes more dialogue which may be short phrases, while obituaries written by New York Times authors are more information-dense and descriptive.
>
> > **how much would you expect generated-texts to approach the human-generated texts by refining the prompt?**
>
> We don’t see an easy way to precisely answer this question. We believe minor modifications of the prompt (“make it as human-like as possible” or variations of that form) will not help. In the limit, a prompt like “Here is a human-written obituary: [...] Please repeat it.” will of course match the human-written text. Providing a content plan matching the human one will give results somewhere in the middle.
>
> >  **could it be that there may be possible segmentations possible and that humans are just more creative in their interpretations?**
>
> Annotators were given unsegmented documents. So, it is possible that annotators may have intuitively grouped sentences together internally while reading for better comprehension - however, we did not ask them to perform segmentation. We saw that different annotators listed slightly different answer sentences, which may have been a result of this. However, we found that LLMs were more reliable at answering questions directly.
>
> > **I'd appreciate some clarification on the statement regarding the exclusion of prior content to keep QUDs free of anaphoric expressions.**
>
> Since this work looks at cross-document answerability of QUDs, it is important for the QUDs to be fully specified and don’t have anaphors (compare: “What is the reason given for the deception?” to “What is the reason for that?”). We found that generating QUDs together with segment decontextualization (Appendices C3 and C4) is an effective way to achieve this.

---

> > ### Comment · Reviewer_UMuD · 2025-06-05
> > **Satisfied with responses**
> >
> > Thank you for your answers. Some points don't really have an answer, but that just reflects the challenges of work to some extent. In any case, you clearly have thought about them and they will be resumed or studied further in future work.

---

### Official Review · Reviewer_kFys · 2025-05-11

**Rating:** 7
**Confidence:** 4
**Ethics Flag:** 1

**Summary:**

The paper presents new similarity metrics for textual fragments based on the QuD methodology. This metrics is used in the evaluation of content generation by LLMs and its comparison with the texts written by humans.

**Questions To Authors:**

How "human" text were collected? It was only mentioned that they were picked from the Reddit. How one can guarantee that these texts were not generated?

What parameteres (top-p, temperature) were used during LLM-based text generation? It can seriously influence model "creativeness".

**Reasons To Accept:**

New original similarity metrics and methodology is proposed and evaluated.
Very extensive evaluation including human validation is performed. Detailed description is provided.
New interesting insights about LLM text generation abilities are provided.

**Reasons To Reject:**

Provided methodology is quite restricted. Proposed measure ignores the order of the textual segments inside the texts and their logical connections. Other discourse theories like PDTB or RST could be more helpful here. Unlike these theories, QuD has no basic framework (segmentation, question generation, question-answer matching), so authors need to constuct it from scratch.
The dataset is very small for such a fine-grained analysis. Only 10 documents written by human are not enough for the established conclusions.
The paper is full of details but some important ones are still missing. Please, see Questions to the authors.

---

> ### Author Response · Authors · 2025-06-02
>
> Thank you for your feedback! Addressing the concerns raised below:
>
> > **Proposed measure ignores the order of the textual segments… Other discourse theories like PDTB or RST could be more helpful here.**
>
> We agree that the lack of a hierarchical structure is a limitation of the current method, though for the ordering of segments, we have studied sequential QUD templates (Section 7). Theoretically, QUD structures can take the form of stacks (Roberts, 2012), trees ([De Kuthy et al., 2018](https://aclanthology.org/L18-1304/)), or dependency structures ([Wu et al., 2023](https://aclanthology.org/2023.emnlp-main.325/), [Ko et al., 2023](https://aclanthology.org/2023.findings-acl.710/)). Yet, as a first step towards quantifying similarity with QUDs, we chose a structure-free (i.e., “flat”) option with a level of abstraction that can be operationalized (lines 80-85). The reason for this is to avoid theoretical complexity that can quickly get out of hand: the structures themselves assume some level of entailment relationship between the questions (in the case of stacks and trees), which is beyond the scope of this paper. (Note that this work already engages QUD with other semantic/pragmatic theories, such as alternative semantics in questions.) Similarities of dependency structures can apply to this work; however, in this paradigm, the QUDs are generated independently from each segment (also see [Wu et al., 2023](https://aclanthology.org/2023.emnlp-main.325/)), thus the structure has no impact on the QUDs themselves but is solely about _which_ sentences invoke the QUD at hand. Since QUDsim measures cross-document QUD answers, we believe that analyzing the QUDs themselves is more meaningful, and leave the structure to future work.
>
> Similarly, it would be great to also use aspects of PDTB and RST, but these too are something we think would be exciting to investigate in the future: the intersection between QUD and coherence relation is a fascinating topic, but not a straightforward one (e.g., see [Riester et al., 2021](https://aclanthology.org/2021.discann-1.5/), [Shahmohammadi et al., 2023](https://aclanthology.org/2023.codi-1.11/), [Ko et al., 2023](https://aclanthology.org/2023.findings-acl.710/), among many others). We intentionally chose the QUD theory because of its nature which bridges the _content_ of discourse with the _structure_ of discourse (for instance, see [Ko et al., 2023](https://aclanthology.org/2023.findings-acl.710/) section 3, where QUDs can be used to characterize coherence relations), which we believe is a solid middle ground that does not drift into similarity strictly in terms of coherence structure. It is indeed very true that QUD is new to the type of investigation here, but this is exactly why we did it!
>
> We will include more discussion around this point in any future version.
>
> > **The dataset is very small for such a fine-grained analysis. Only 10 documents written by human are not enough for the established conclusions.**
>
> The focus of this work is a similarity measure, for which we have 100 documents in total. Most of these documents are from different LLMs, for us to draw conclusions about templaticness _across_ LLMs. Our goal is not to study human-LLM divergences or establish the higher creativity of humans, which has been documented in other work ([Padmakumar and He 2024](https://arxiv.org/abs/2309.05196), Chakrabarty et al., 2024, Lu et al., 2025), so more examples are not needed to establish the conclusions of this work.
>
> > **How "human" text were collected?…How one can guarantee that these texts were not generated?**
>
> The Reddit responses are part of the [WritingPrompts](https://aclanthology.org/P18-1082/) dataset that was introduced in 2018 by Fan et. al. The models predating 2018 were unable to produce long coherent discourse that constitutes the highly involved stories we collected and studied. The obituaries were from New York Times writers, who we are assuming to be credible human authors.
>
> > **What parameters (top-p, temperature) were used during LLM-based text generation? It can seriously influence model "creativeness".**
>
> We use the default parameters (temperature=1 and top-p=1). We included these details in Appendix B. We experimented with different temperatures in preliminary experiments, but did not observe meaningful differences. This is in line with prior work that found that temperature variation does not reduce the templatic nature of LLM outputs, even at the POS/lexical level (Shaib et al., 2024). In other experiments, we have found that writing quality starts to degrade according to reward model scores when higher temperatures (above 1.3) are used.

---

> > ### Comment · Reviewer_kFys · 2025-06-06
> >
> > Thanks for your answers. I will increase my score.

---

### Official Review · Reviewer_HZh2 · 2025-05-12

**Rating:** 8
**Confidence:** 5
**Ethics Flag:** 1

**Summary:**

This paper is motivated by the intuition that LLMs often generate "boring" text and begins to evaluate discourse-level similarities in LLM-generated texts. The authors propose a new metric, called QUDSim, based on a discourse theory: Questions Under Discussion (QUD). The evaluation is extensive and comprehensive.
I believe we should welcome more computational linguistics (CL)-inspired work in COLM. I vote positively now.

**Questions To Authors:**

**Q1:** I have a question regarding reversed questions in discourse and their impact on QUDSim calculation. Suppose $Q_{s,i}$ and $Q_{s,j}$ form an answer to a question $q$, and $Q_{t,i}$ and $Q_{t,j}​$ form another question $q'$, and $q$ and $q'$ happen to be reverses of each other (e.g., reason vs. result). Would this result in a high QUDSim score? If $q'$ changes its reverse form and becomes identical to $q$, would the QUDSim score increase?

**Q2:** Do different types of questions (and the underlying discourse dependencies) affect the QUDSim score differently? That is, perhaps a "What is the reason for..." question contributes more heavily than a "What describes..." question, since the former introduces causality and may hypothetically be more important than the latter, which might merely paraphrase.

**Q3:** Can you provide a taxonomy of the questions used? Thank you for the comprehensive appendices, but I couldn't find this in the provided material (please point it out if I missed it).

## Recommended Citations

There are recent methods with similar aims. I recommend the authors review and cite the following:

- [_QUDeval: The Evaluation of Questions Under Discussion Discourse Parsing_](https://aclanthology.org/2023.emnlp-main.325.pdf), EMNLP 2023
- [_Discursive Socratic Questioning: Evaluating the Faithfulness of Language Models' Understanding of Discourse Relations_](https://aclanthology.org/2024.acl-long.341/), ACL 2024

These prior works employ (multi) question answering (similar to your approach) to evaluate how well models understand discourse structures or relations. The latter includes a sequence of question generation and answering steps, which is similar in spirit to the pipeline in this manuscript.

**Reasons To Accept:**

The paper has the following strengths (S):

**S1: Discourse theory**
The paper applies the discourse theory QUD to the evaluation of discourse structure similarity. The citations and discussion (around Lines 94-108) are particularly insightful. The proposed metric is a simple average over answerability, which appears simple and elegant.

**S2: Evaluation appears comprehensive and well-defended**
The paper evaluates a variety of models and presents many interesting findings. I especially appreciate the findings in Section 6, including that LLMs tend not to employ diverse discourse structures and that cross-LLM comparisons also yield similar structures.

**S3: The presentation of the paper is excellent**
The notations and symbols are clear and necessary. The authors wisely allocate space for natural language examples, which is important for a CL-oriented paper. The figures are clear and easy to follow.

**Reasons To Reject:**

The paper has the following weaknesses (W):

**W1: Why not simply measure tree similarity?**
As we know, discourse structure can be represented as a tree. Following the notations in the paper, we could create links between $D_{s,i}$, $D_{s,j}$  (and likewise $D_{t,i}$ and $D_{t,j}$). These links between different nodes (segments) form the tree structure of the discourse. There is prior work, especially from before the LLM era, on measuring tree similarity (e.g., [_Sentence Similarity based on Dependency Tree Kernels for Multi-document Summarization_](https://aclanthology.org/L16-1452), LREC '16). I wonder whether it would be possible to set up a baseline using such methods. These offer simple and straightforward realizations of the paper's idea. Please correct me if this is out of scope :)

This leads to another limitation of the method: QUDSim is linear (by averaging over answerability), while discourse structure is hierarchical. I wonder whether both local and global similarities are taken into account? Also, do different question types carry different weights in the QUDSim score? (See the question section below for details.) Do recursive structures compute weights recursively?

**W2: Correlation with other metrics?**
As a newly proposed metric, QUDSim would benefit from further justification if it is to be adopted widely.

As with the proposal of ROUGE ([_ROUGE: A Package for Automatic Evaluation of Summaries_](https://aclanthology.org/W04-1013.pdf)), evaluation against human judgments increases a metric's trustworthiness. The paper currently includes some LLM-as-judge results, but I wonder whether any human validation has been conducted for QUDSim. I think Section 5.1 satisfies this by checking question answerability, which is acceptable. However, I wonder if there are any more "end-to-end" alignment evaluations for the final QUDSim score?

I also believe it would be valuable to report correlations between QUDSim and other established scores (e.g., ROUGE, METEOR, or BERTScore). It is fine if QUDSim diverges from them, but it would be interesting to show where it aligns with those metrics, and where it doesn't (e.g. the "hard" samples can be better aligned with QUDSim because it goes deeper into the structure?).

A few more details: Do you provide an upper bound (u), lower bound (l), or chance estimate (c) for QUDSim? Do these values vary with the length of the discourse? I believe such information should be included as part of the standard specification.

---

> ### Author Response · Authors · 2025-06-02
>
> Thank you for your detailed feedback and comments! Addressing the concerns raised below:
>
> > **W1: Why not simply measure tree similarity?**
>
> In QUD dependency trees ([Wu et al., 2023](https://aclanthology.org/2023.emnlp-main.325/), [Ko et al., 2023](https://aclanthology.org/2023.findings-acl.710/)), the edge labels are the questions themselves: the edge goes from the answer to the QUD to where the QUD is evoked (or anchored) in prior context. The core idea of QUDsim is to measure cross-document QUD answers, which ultimately comes down to analyzing the QUDs themselves, i.e., the edges of the dependency structures. Only comparing the structures themselves, without the QUDs, will not give us the type of similarity we are looking for.
>
> We do study some notion of global similarity by analyzing _ordered_ QUDs (Section 7). This analysis reveals the existence of templates consisting of QUD sequences. Theoretically, QUD structures can take the form of stacks (Roberts, 2012), trees ([De Kuthy et al., 2018](https://aclanthology.org/L18-1304/)), or dependency structures ([Wu et al., 2023](https://aclanthology.org/2023.emnlp-main.325/), [Ko et al., 2023](https://aclanthology.org/2023.findings-acl.710/)). Yet, as a first step towards quantifying similarity with QUDs, we chose a structure-free (i.e., “flat”) option with a level of abstraction that can be operationalized (lines 80-85). The reason for this is to avoid theoretical complexity that can quickly get out of hand: the structures themselves assume some level of entailment relationship between the questions (in the case of stacks and trees), which is beyond the scope of this paper.
>
> > **W2: Correlation with other metrics?**
>
> This work comparatively analyzes QUDsim with three other types of similarity measures which we believe are representative of their kind: (1) N-gram overlap based metrics, from 1-4grams; (2) embedding-based metric, with cosine; (3) LLM-based similarity (see Section 4, “baselines”). ROUGE and METEOR are N-gram based; ROUGE results are worse than 1-4gram baselines included in this work (Rouge-L achieved an F1 score of only 0.026 when computed in the same manner as other values presented in Table 2), thus we did not include it, but we are happy to add to the table. BERTScore is embedding-based, thus will not have divergent results from type (2). (Note that BERTScore itself is highly uninterpretable, with highly clustered numerical values). Our preliminary investigation showed that this is indeed the case, and we can add this to the appendix.
>
>
> >  **Do you provide an upper bound (u), lower bound (l), or chance estimate (c) for QUDSim? Do these values vary with the length of the discourse?**
>
> The upper bound of QUDsim is 1, i.e., when comparing the same document with itself, all answerability scores will be 1. This is shown in the diagonal values of Figure 5. The lower bound of QUDsim is 0, i.e., when no source QUD gets answered in a target document and vice versa. Random pairs of documents in creative writing will also receive values close to 0 since there is little chance that they share the same QUDs; we validated that this is indeed the case with LLM-generated texts randomly sampled from different story prompts. For  LLM-generated obituaries of different people, Figure 4 (left) shows the QUDsim values.
>
> > **Q1: Reverse questions and QUDsim:**
>
> This is a very interesting question! Assume that q takes the form “What is the reason for X” and q’ takes the form “What is the result of X”. This means that Q(S,i) and Q(S,j) state the reason for X whereas Q(t,i) and Q(t,j) state the result of X. Since these are different questions, the answerability scores for q in Q(t,i) and Q(t,j) will be 0, and vice versa; thus, there should be no increase in QUDsim. This is a very compelling angle of analysis though, which also speaks to connecting this framework with coherence discourse, and we’d be happy to explore further.
>
> > **Q2: Do different types of questions (and the underlying discourse dependencies) affect the QUDSim score differently?**
>
> We have not conducted an analysis from this angle. Similar to the question above, we also find it interesting, and will be happy to explore further in future work.
>
> > **Q3: Can you provide a taxonomy of the questions used?**
>
> Yes, we would be happy to run the question type classifier from [Cao and Wang (2021)](https://aclanthology.org/2021.acl-long.502/) and provide the frequency of questions in each type, similar to what was done in Ko et al., 2022.
>
> > **Suggested citations:**
>
> Thanks for the pointers. We will include these in any future version.

---

> > ### Comment · Reviewer_HZh2 · 2025-06-04
> > **Thanks for your response & Raise the score**
> >
> > Hi authors,
> >
> > Thank you for the detailed responses. I find your responses to W2, Q1, Q2, and Q3 and particularly substantial. I will raise my score. I look forward to your additions in the final version :)
> >
> > Best, HZh2

---

### Official Review · Reviewer_biQ6 · 2025-05-12

**Rating:** 6
**Confidence:** 4
**Ethics Flag:** 1

**Summary:**

This paper addresses the problem of measuring the structural similarity between two piece of long-form texts.

Based on the theory of Questions Under Discussion (QUD), which is a model of discourse that views the progression of a narrative as answers to a series of implicit questions, this paper first propose a way to quantify the answerability of a target document to a source document. Then the paper propose to measure the structural similarity QUDsim as the harmonic mean between the bidirectional answerability.

This paper builds a dataset by collecting responses from GPT, Claude and Gemini and human for prompts ranging from three domains, and then conduct analysis on the structural similarity of the texts generated by different sources. The proposed method outperform a range of baseline methods on human evaluation.

**Questions To Authors:**

- In Figure 2, some annotations are not explained, such as $A_{Q_k}$.

- Missing reference: Previous works on measuring functional discourse structure similarity: https://aclanthology.org/2024.naacl-short.9/

**Reasons To Accept:**

- This paper is well written and complete in structure. It has clear motivation, good theoretical background and clear evaluation and analysis.

- The analysis shows some inspiring results on the structures of the texts generated by latest LLMs.

**Reasons To Reject:**

- How do you know where to split the document into segments?

- The granularity of questions in the QUD framework is tricky. If the question is very specific, then the content between source doc and target doc need to be very similar. However, how to quantify or set this granularity is undiscussed.

---

> ### Author Response · Authors · 2025-06-02
>
> Thank you for your feedback and comments! Addressing some of the concerns below:
>
> > **How do you know where to split the document into segments?**
>
> We follow existing text segmentation principles in the LLM-based text segmentation process: “each chunk is about one atomic topic” (Appendix C.2). That said, a first-principles definition for segmentation itself is an open research area ([Retkowski and Waibel, 2024](https://aclanthology.org/2024.eacl-long.25.pdf) ) In this work specifically, we made the assumption (which was indeed the case from observations) that by using GPT-4o for all queries throughout the pipeline, the LLM makes judgments that are somewhat consistent, so its understanding of granularity may not drastically vary from one document to the next. A full analysis of how granularity impacts similarity is left for future work. Our intuition is that as long as the granularity is fine enough (e.g., paragraph level or a bit below), the relative similarity judgments of pairs of texts here won’t change much.
>
> >  **The granularity of questions in the QUD framework is tricky. [...] how to quantify or set this granularity is undiscussed.**
>
> Indeed, QUDs can come at different levels of abstraction. Since our QUDs correspond to the segments (1-2 QUDs per segment, Table 1), which are fine-grained, we avoid having very high-level QUDs. Most QUDs can be answered in just a few sentences, since each segment is at the subparagraph level.
>
> Additionally, our use of entity abstraction helps to decouple fine-grained information from the QUDs used to find similarity. This processing step further reduces the dependence on the precise similarity level. The intrinsic evaluation of this abstraction is in Appendix D.2.
>
> > **In Figure 2, some annotations are not explained, such as A_Q_k.**
>
> Q is the set of all QUDs generated for a given document. Q_k is the k^th QUD in this set. So, A_Q_k is the set of answer sentences for the k^th QUD. We will make this more clear in any future version.
>
> > **Missing Reference**
>
> Thank you! We will include this.

---

> > ### Comment · Reviewer_biQ6 · 2025-06-11
> >
> > Thank you for your response.
> >
> > You've addressed my questions, and I don't have any further concerns.

---

### Decision · Program_Chairs · 2025-07-08

**Decision:**

Accept

**Comment:**

This paper is about measuring discourse-level similarity between LLM generated texts. The approach is based on Questions Under Discussion, theory under pragmatics governing the interpretation and future utterances/text in a discourse.

The problem is a worthwhile one. As models become good, there is a need to identify the important opportunities for improvement. This paper appears to go in that direction empirically showing that LLMs reuse discourse structures very often and diverge from human authors.

All the reviewers are in consensus about the quality of the work, its framing, claims, choices and evaluation. I recommend this paper for publication. I also suggest an oral presentation.